# Spa Therapy Efficacy in Mental Health and Sleep Quality Disorders in Patients with a History of COVID-19: A Comparative Study

**DOI:** 10.3390/diseases12100232

**Published:** 2024-10-01

**Authors:** Maria Costantino, Valentina Giudice, Francesco Marongiu, Mariagrazia Bathilde Marongiu, Amelia Filippelli, Horst Kunhardt

**Affiliations:** 1Department of Medicine, Surgery, and Dentistry, University of Salerno, 84081 Baronissi, Italy; vgiudice@unisa.it (V.G.); afilippelli@unisa.it (A.F.); 2Non-Profit Association F.I.R.S.Thermae (Interdisciplinary Training, Researches and Spa Sciences), 80078 Pozzuoli, Italy; 3Department of Women, Child and General and Specialized Surgery, University “Luigi Vanvitelli”, 80138 Naples, Italy; 4Faculty Applied Healthcare, Deggendorf Institute of Technology–University of Applied Sciences, 94469 Deggendorf, Germany; horst.kunhardt@th-deg.de

**Keywords:** COVID-19, spa therapy, depression, anxiety, sleep disorders

## Abstract

The COVID-19 pandemic has left behind mental health issues like anxiety, depression, and sleep disorders among survivors. This study assessed the efficacy of spa therapy in enhancing psychological well-being and sleep quality in individuals with chronic arthro-rheumatic, respiratory, and otorhinolaryngological diseases, including COVID-19 recoverees. Our prospective observational study included 144 Caucasian subjects from three Italian spas who underwent a 2-week spa therapy cycle, involving balneotherapy and/or inhalation treatments. Symptoms were assessed with the Visual Analogue Scale (VAS), psychological well-being with Depression Anxiety Stress Scales-21 items (DASS-21), and sleep quality with the Insomnia Severity Index (ISI). Significant reductions in VAS scores for arthro-rheumatic, respiratory, and otorhinolaryngological symptoms were observed after spa therapy, as well as for DASS-21 and ISI scores for sleep quality, transitioning to less severe insomnia categories. Females had more pronounced improvements in DASS-21 scores and sleep quality. Subjects with and without prior SARS-CoV-2 infection experienced significant reductions in anxiety, depression, and stress, with more pronounced improvements in those without prior infection. COVID-19 survivors also showed significant ISI score improvements. Spa therapy is a promising complementary treatment for improving mental health and sleep quality in chronic disease patients, including COVID-19 survivors.

## 1. Introduction

The COVID-19 pandemic has posed an unprecedented challenge to global public health, and beyond its direct physical effects, its long-term impact on mental health and psychological well-being in affected individuals is becoming increasingly evident [1]. Among the challenges faced by COVID-19 survivors, anxiety, depression, and sleep disorders emerge as widespread and persistent problems [1,2,3,4,5], as it is estimated that over a third of survivors develop psychiatric disorders within six months of recovery [2]. In addition, clinically significant post-traumatic stress disorder, anxiety, and/or depression are observed after 14–90 days from diagnosis [1], and subjects also report significant alterations in sleep quality, ultimately leading to poor overall health and post-illness recovery [5]. In response to these challenges, *Salus per aquam* (Spa) therapies are emerging as a potential complementary therapeutic approach to address anxiety and depression, and to improve sleep [6,7,8].

Since the Romans, spa therapies have been used for treating various medical conditions because of their healing, anti-inflammatory, antioxidant, and relaxing properties [8,9,10,11]. Indeed, inflammation is one of the principal pathophysiological mechanisms in multiple chronic conditions, and it can be targeted by spa interventions that can modulate inflammatory responses and reduce oxidative stress. For example, inflammation during airway infections and the recurrence of infectious airway diseases are significantly decreased after spa inhalation treatment [12,13,14], as well as tumor necrosis factor alpha (TNFα) and calprotectin levels in nasal mucosa [15], or interleukin (IL)-1 and IL-6 levels in nasal washes chronic obstructive pulmonary disease [11,16]. Moreover, IL-2, IL-4, and interferon-gamma (IFN-γ) levels that rise during upper respiratory tract infections are reduced after spa inhalation treatment, while anti-inflammatory cytokines, such as IL-10, are increased [16,17,18]. Spa inhalation therapies induce the amelioration of inflammation and mucolytic actions, leading to improvements in normal acoustic curve functions, with enhanced hearing in chronic upper-airway and middle-ear inflammation [19], or the restoration of altered olfactory function in subjects with chronic rhinosinusitis [20]. After spa balneotherapy or mud balneotherapy, circulating prostaglandin E2, leukotriene B4, and TNFα are also reduced in patients with osteoarthritis or fibromyalgia [21,22,23,24], as well as redox status after spa therapy for musculoskeletal [25,26] and skin disorders [27]. Despite these well-known beneficial effects of Spa therapy in inflammatory conditions, research on its effectiveness for mental health disorder treatment, such as anxiety, depression and sleep disorders, is limited. Therefore, a heterogeneous study population with different types of chronic inflammatory conditions, such as arthro-rheumatic, respiratory, and otorhinolaryngological diseases, as well as COVID-19 survivors, was included in our investigation to assess the efficacy of Spa treatments on mitigating inflammation, regardless of underlying diseases, and to explore cross-sectional effects on various conditions.

In this study, we aim to examine the efficacy of Spa therapy for well-being and mental health recovery in subjects with chronic inflammatory conditions and in COVID-19 survivors, thus proposing spa treatments as a complementary and effective tool for improving the mental health of this vulnerable population. Our results were also stratified by gender to identify different impacts on specific dimensions of psychological well-being, including anxiety, depression, stress, and sleep disorders, as these aspects are often altered in individuals with chronic conditions, and spa therapy can relieve stress and improve mood and overall mental health.

## 2. Materials and Methods

### 2.1. Population and Study Design

A total of 144 Caucasian subjects were included in this study and enrolled from three Italian spas: “Stufe di Nerone” Baths (Bacoli—Naples), “Giardini Poseidon” Baths (Ischia—Naples), and “Terme dei Colli Asolani” Baths (Pieve del Grappa, Crespano, Treviso). These subjects had a diagnosis of chronic arthro-rheumatic, and/or respiratory, and/or otorhinolaryngological diseases, and received a prescription from a general practitioner or specialist for a 2-week therapeutic spa cycle (balneotherapy and/or inhalation). This study was approved by our local Ethics Committee “Campania Sud”, Naples, Italy (approval n. 44 r.p.s.o./2023 on 7 March 2023) and conducted in accordance with the Declaration of Helsinki. A specific case report form (CRF) was used to collect patients’ data. Participant demographics, inclusion and exclusion criteria, therapeutic interventions, treatment protocols, outcome measures, and data collection time points are summarized in Table 1.

### 2.2. Outcomes

At baseline and after 2 weeks of spa therapy, the enrolled subjects were evaluated for the following outcomes: the impact of spa treatment on symptoms was investigated using the Visual Analogue Scale (VAS), with a score ranging from 0 (“no symptom”) to 4 (“symptom as bad as it could possibly be”), while its impact on specific aspects of psychological well-being (e.g., anxiety, depression, and stress) was assessed by a valid, quick, and reliable tool, the Italian version of the psychometric questionnaire “Depression Anxiety Stress Scales-21 items” (DASS-21) [28]. This questionnaire was completed the week preceding spa completion, and consists of three subscales (depression, anxiety, and stress), each containing 7 items. Respondents rated the intensity of their symptoms on a four-point scale from 0 (does not apply to me at all) to 3 (applies to me very much or most of the time). The DASS-21 severity categories are: normal (scores: 0–9 for depression, 0–7 for anxiety, and 0–14 for stress), mild (scores: 10–13 for depression, 8–9 for anxiety, and 15–18 for stress), moderate (scores: 14–20 for depression, 10–14 for anxiety, and 19–25 for stress), severe (scores: 21–27 for depression, 15–19 for anxiety, and 26–33 for stress), and extremely severe (scores: ≥28 for depression, ≥20 for anxiety, and ≥34 for stress).

Sleep quality was measured using the Insomnia Severity Index (ISI) questionnaire [29,30,31], a reliable and valid tool for identifying cases of insomnia in the general population and for quantifying severity of insomnia symptoms, sleep satisfaction, and impact of sleep on daytime functioning. Thus, this index was also employed for the evaluation of treatment efficacy. This self-assessment questionnaire is composed of 7 items and was conducted the week preceding spa completion. Each item is rated on a 5-point scale (from 0 to 4). The total scores range from 0 to 28, and a total score between 0 and 7 indicates “absence of clinically significant insomnia”, while a score between 8 and 14 suggests “subthreshold insomnia” (a clinical form of insomnia that, although not meeting all diagnostic criteria, causes a significant reduction in the individual’s social functioning, manifesting with fewer and less intense symptoms than usual but still negatively impacting quality of life). A score between 15 and 21 indicates the presence of “moderate clinical insomnia”, and a score between 22 and 28 suggests “severe clinical insomnia”. The occurrence of undesired events was recorded during the entire study period.

### 2.3. Data Analysis

Characteristics of the study population were assessed by descriptive analysis. Continuous variables were presented as mean ± standard deviation (SD), and the two-group comparison was performed using *t*-test for normally distributed data, while Wilcoxon’s signed-rank test or Mann–Whitney test was for non-normally distributed variables. The percent variation (Δ%) in the total score of DASS-21 or other values before and after spa therapy was calculated as follows: Δ% = [(total-score after–total-score before)/total-score before] × 100. Chi-square test was applied where appropriate. A *p* value < 0.05 was considered statistically significant. Post-hoc power analysis was performed using G*Power software (v.3.1.9.6; Franz Faul, Universitat Kiel, Kiel, Germany), and for before–after spa therapy, using the total cohort of 144 patients; the power was 98%, while for comparisons between females (n = 98) and males (n = 46), the power was 87%. Data collection and analysis were carried out using the SPSS 23.0 statistics package.

## 3. Results

### 3.1. Impact of Spa Treatment on Disease-Related Symptoms and Mental Health

All enrolled subjects completed the spa cycle. VAS scores related to arthro-rheumatic, respiratory, and otorhinolaryngological symptoms were measured at the end of the Spa cycle, and we showed a significant reduction compared to baseline (*p* < 0.05) (Table 2), as well as for DASS-21 scores (Table 3). At the end of the spa cycle, the percentage of patients with total scores within the normal range for depression significantly increased (from 46.5% to 67.4%, *p* < 0.01), as well as for anxiety (from 43.1% to 75.7%, *p* < 0.01), and stress (from 50.7% to 82.6%, *p* < 0.01) (Table 3), while no variations were observed for mild anxiety and depression categories (Table 3).

In our cohort, 27% (n = 39) of individuals reported sleep disorders at baseline, and they had a mean age ± SD, 53 ± 16 years old (median age, 58 years; range, 26–78) and a mean BMI (Kg/m^2^) ± SD of 24 ± 3.8, and were mostly females (n = 25, 64%). At baseline, 36% (n = 14) of patients had a total ISI-score indicating “no clinically significant insomnia”, with a mean total score of 3.4 ± 2.2, and no variations were observed at the end of Spa cycle in this group. The remaining 64% (n = 25) of subjects exhibited various severity categories of insomnia at baseline, with a mean total ISI score of 15.0 ± 5.0, corresponding to “moderate-severity clinical insomnia”.

Following the spa cycle, a significant reduction in mean total ISI score was observed (7.0 ± 4.0; *p* < 0.01), with a transition to the category of “no clinically significant insomnia”. Next, we compared each ISI score severity category before and after spa therapy, and we showed that subjects presenting with “no clinically significant insomnia” (36%; n = 14) did not show significant variations at the end of the spa cycle (3.4 ± 2.2 vs. 2.6 ± 2.0; *p* > 0.05), while individuals with “sub-threshold insomnia” (33.3%; n = 13) displayed significant amelioration at the end of the spa therapy (11.2 ± 2.2 vs. 7.1 ± 4.2; *p* < 0.05) with a shift towards the category of “no clinically significant insomnia”, and patients with “clinical insomnia moderate severity” at baseline (23%; n = 9) were markedly improved at the end of treatment (17.7 ± 2.1 vs. 6.3 ± 3.2; *p* < 0.02). Only three subjects (7.7%) had “clinical insomnia severe” at baseline, and a reduction in the mean total score was observed (24.0 ± 1.7 vs. 9.0 ± 6.6); however, due to the small sample size, statistical analysis could not be performed (Table 4).

### 3.2. Females Suffer More Frequently from Sleep Disorders

To examine gender differences in response to spa therapy, we stratified the enrolled subjects by sex, and the majority of individuals were females (n = 98; 68%; mean age ± SD, 57 ± 13.7 years; range, 24–78 years; median age 59 years; BMI [Kg/m^2^] ± SD, 24.8 ± 4.1), while 32% of patients (n = 46; 32%; mean age ± SD, 54 ± 16.5 years; range, 25–77 years; median age, 57 years; mean BMI [Kg/m^2^] ± SD, 25.9 ± 4.0) were males. At baseline, no significant differences were described in mean total DASS-21 score between sexes; conversely, although Spa therapy produced a significant improvement in mental health conditions in both sexes (*p* < 0.01), this improvement was greater in females compared to males (Table 5).

Similarly, females more frequently complained of sleep disorders (n = 25; 64%; mean age ± SD, 53 ± 15 years; range, 27–78 years; median age, 58 years; mean BMI [Kg/m^2^] ± SD, 23 ± 3.5) compared to males at baseline (n = 14; 36%; mean age ± SD, 51 ± 17 years; range, 26–71 years; median age, 56 years; mean BMI [Kg/m^2^] ± SD, 25 ± 4.1), with significantly higher mean total ISI score in females compared to males (*p* < 0.05) (Table 5). At the end of the spa cycle, a reduction in ISI score was observed in both sexes and was significantly greater in females compared to males (Table 5).

### 3.3. Impact of Previous SARS-CoV-2 Infection on Mental Health

Finally, enrolled subjects were stratified by at least one previous SARS-CoV-2 infection, and most patients (n = 107) had contracted COVID-19 confirmed by molecular test (mean age ± SD, 54 ± 15 years; range, 24–78 years; median age, 57 years; female 72% and male 28%; mean BMI [Kg/m^2^] ± SD, 25 ± 4.2). In detail, 68% of them (n = 73) reported mild COVID-19 treated at home, while 32% (n = 34) had moderate/severe forms, with only one patient requiring oxygen support, although no subjects were admitted to intensive care units. The average symptom duration was 9 ± 1.2 days for mild and 13 ± 2.2 days for moderate/severe cases. Moreover, 65% (n = 70) of those subjects who had a previous COVID-19 infection reported long-term sequelae.

In contrast, the remaining subjects (n = 37; mean age ± SD, 62 ± 12 years; range, 36–76 years; median age, 66 years; female 57% and male 43%; mean BMI [Kg/m^2^] ± SD, 25 ± 3.7) had never contracted SARS-CoV-2 infection. Spa therapy significantly improved symptoms in both groups (Table 6), with reduction in anxiety, depression, and stress after spa cycle (*p* < 0.01) (Table 7).

This improvement was more pronounced in subjects who had never contracted COVID-19, while individuals with a history of COVID-19 infection had higher scores of anxieties, stress, and depression at baseline, indicating a persistent impact of SARS-CoV-2 infection on mental health. However, also in this group, a significant reduction in symptoms was observed after spa therapy (Table 7). Among subjects with sleep disorders at baseline (n = 39), 74% of them (n = 29; mean age ± SD, 51 ± 16 years; range, 26–78 years; median age, 55 years; 79% female and 21% male; mean BMI [Kg/m^2^] ± SD, 23 ± 4.1) had a previous SARS-CoV-2 infection, while the remaining 26% of patients (n = 10; mean age ± SD, 58.4 ± 13 years; range, 38–71 years; median age, 65.5; 20% female and 80% male; mean BMI [Kg/m^2^] ± SD, 25.6 ± 2.6) had never contracted COVID-19. At baseline, the mean total ISI score was higher in subgroup with a previous COVID-19 infection (12.1 ± 7.2, “sub-threshold insomnia” category), and was markedly reduced after treatment (*p* < 0.01), shifting to the “no clinically significant insomnia” category (Table 7). In subjects who had never contracted COVID-19 infection, no sleep disorders at baseline (7.4 ± 5.4) and at the end of spa therapy were observed (Table 7).

## 4. Discussion

Chronic disorders, such as arthro-rheumatic, respiratory, and otorhinolaryngological diseases, not only influence the physical well-being of affected patients, but are also often associated with emotional and psychological alterations. Anxiety, depression, and stress are common symptoms that negatively affect their quality of life, including sleep quality. In our prospective observational study, we demonstrated that spa therapy, both balneotherapy and inhalation therapy, can positively impact the psychological well-being and sleep quality of individuals suffering from these conditions, including those with long-term psychological consequences of COVID-19 infection, even after recovery from the acute phase and having a negative molecular swab test [32,33,34,35,36]. Published studies report that 53.8% of individuals experience moderate to severe psychological symptoms, with 28.8% of anxiety and 16.5% of depression [33], as well as reduced sleep quality in Long-COVID populations, especially women, highlighting the need for careful monitoring and appropriate treatments [34]. Possible mechanisms underlying psychiatric sequelae in COVID-19 infection include neurotropism, immune response to SARS-CoV-2, hyper-reactivity of the hypothalamic–pituitary–adrenal axis, neuroinflammation, and neuronal death [1].

Spa mineral waters have anti-inflammatory, analgesic, and decongestant properties, as well as a relaxing environment and positive social interactions, as already described in previous in vitro studies and clinical trials, such as that demonstrating the antioxidant effects of Portuguese sulfurous/bicarbonate/sodic spa mineral water mediated by increased inducible nitric oxide synthase expression [37,38,39,40]. Therefore, spa therapy can be considered an effective alternative strategy for improving psychological well-being in patients with chronic arthro-rheumatic, respiratory, and otorhinolaryngological diseases. In our study, we observed a significant reduction in mean total DASS-21 scores (anxiety, depression, and stress) and in most of physical symptoms detected at baseline, with an increase in the percentage of subjects with normal scores for anxiety (+74%), depression (+45%), and stress (+63%). Similarly, our results highlight the significant improvements in sleep disorders after treatment, with most subjects transitioning to less severe categories of insomnia, indicating that spa therapy could be particularly effective for reducing insomnia severity. This effect can be caused by synergistic effects of muscle relaxation and stress reduction induced by Spa treatment.

During balneotherapy cycles, cortisol, a biomarker of stress, decreases, blood circulation and tissue reoxygenation increase, anti-inflammatory and muscle-relaxant actions are observed, and a reduction in sleep onset latency with a significant improvement in sleep quality is also reported [7,10,41,42,43,44,45]. Treatments with the mineral elements and chemical compounds in mineral waters, as well as the peloids/muds used in balneotherapy, are associated with greater and continuous symptom relief in osteoarthritis, chronic low-back pain, and rheumatoid arthritis, compared to “non-mineral” treatments [46]. Several studies have highlighted the benefits in treating chronic respiratory disease of inhaled salsobromoiodic–sulfate mineral waters, such as those used in our study, because of the presence of bioactive mineral components [15,47,48,49,50,51]. For example, sodium chloride acts as an antiseptic, bromine and calcium have analgesic and sedative effects on respiratory mucosa, and sulfur stimulates IgA secretion, improving immune responses in the respiratory tract. Moreover, sulfates in Spa waters have mucolytic and relaxing actions on tracheobronchial smooth muscle, while magnesium helps reduce bronchial hyperreactivity. Finally, these mineral waters can improve ciliary motility, which is particularly useful for those suffering from chronic rhinosinusitis.

Females are more likely to suffer from mental health disorders compared to males, because of a higher burden and an increased susceptibility, with a global annual prevalence of depression of 5.5% in females and 3.2% in males, with a 1.7-fold greater incidence in women, as also documented by several World Health Organization reports [52,53,54,55]. In our study, we confirmed that females were more affected by psychological disorders than men, and although spa therapy produced significant improvements in both sexes, females showed more pronounced improvements in DASS-21 and ISI scores. Females were more likely to have “sub-threshold insomnia” at baseline, which decreased to “absence of clinically significant insomnia” at the end of spa cycle. Conversely, males did not show sleep disorders at baseline and at the end of treatment, reflecting a greater vulnerability of women to sleep disorders, while also a better responsiveness to spa therapy. COVID-19 is associated with a greater risk of post-traumatic stress disorder, and various mental health conditions, including anxiety and insomnia [1]. Indeed, when stratified by previous SARS-CoV-2 infection, our patients with a history of COVID-19 had higher DASS-21 and ISI scores at baseline compared to those subjects who had never contracted the infection, suggesting that this condition might persist in patients, potentially as Long-COVID syndrome, significantly worsening quality of life and psychological well-being [1,2]. Moreover, despite the spa therapy reducing symptoms in both groups, subjects with a history of COVID-19 experienced a less effective improvement compared to those who had never contracted the virus. Similar trends were also observed for sleep disorders, with a significant amelioration after the spa cycle. These results could be related to the positive anti-inflammatory, antioxidant, and muscle-relaxing effects of the natural mineral spa waters used, including salsobromoiodic–sulfate and bicarbonate waters, as well as the spa methods employed, such as balneotherapy and/or inhalation therapy. These effects, especially the relaxing properties, could also explain clinical improvements in psychological well-being caused by muscle tension and stress level relief in the context of sleep disorders and anxiety [56], further suggesting spa therapy as a complementary treatment for chronic conditions.

Our study has several limitations: (i) its observational nature that did not allow us to establish a direct causal relationship between spa therapy and observed improvements; (ii) the absence of a control group; and (iii) the small sample size for certain categories that did not permit specific comparisons, such as those related to insomnia severity categories. Therefore, further research with controlled and randomized designs is needed to confirm these findings.

## 5. Conclusions

In conclusion, our findings suggest that spa therapy could be a valuable complementary therapeutic option for improving psychological well-being and sleep quality in patients with chronic conditions, including those with a history of COVID-19 infection. However, further controlled and randomized studies are needed to confirm these findings.

## Figures and Tables

**Table 1 diseases-12-00232-t001:** Study design and population.

Feature	Description
Type	Prospective observational study
Participants	144 Caucasian subjects
Sex	68% female, 32% male
Age (Mean ± SD)	56 ± 15 years (range: 24–78 years)
BMI (Mean ± SD)	25 ± 4.0 kg/m^2^
Inclusion criteria	Age ≥ 18 years Diagnosis of chronic arthro-rheumatic, respiratory, and/or otorhinolaryngological diseases Written informed consent
Exclusion criteria	Acute clinical conditions Cancer and autoimmune diseases Absence of signed written informed consent
Diseases, n (%)	Chronic arthro-rheumatic diseases, 106 (73.6) Respiratory diseases, 31 (21.5) Otorhinolaryngological diseases, 7 (4.9)
Duration	2-week spa cycle (balneotherapy and/or inhalation therapy)
Protocols	Balneotherapy Full-body immersion (excluding head) in spa mineral waters at 37–38 °C for 20 min. 30 min rest after treatment. 12-day complete cycle, consisting of 12 consecutive baths (once daily) in a single tub, preferably in the morning and under fasting conditions. For chronic arthro-rheumatic diseases. Inhalation Crenotherapy Twelve 10 min applications of spa mineral waters as direct jet and 12 aerosols. 10 min rest between applications. For chronic inflammation and/or irritation of upper and lower respiratory tracts and otolaryngological diseases.
Outcome measures	Symptoms: Visual Analogue Scale (VAS). Psychological well-being: Depression Anxiety Stress Scales-21 items (DASS-21) Sleep quality: Insomnia Severity Index (ISI).
Data collection	At baseline and after 2 weeks of spa therapy

**Table 2 diseases-12-00232-t002:** Impact of spa therapy on chronic disease-related symptoms (n = 144).

Symptoms, (n)	Before Spa Mean VAS ± SD	After Spa Mean VAS ± SD	*p* Value
Arthro-rheumatic Symptoms			
Pain in functional performance (93)	3.4 ± 0.6	2.0 ± 1.1	<0.01
Morning stiffness (85)	3.2 ± 0.8	1.7 ± 1.2	<0.01
Paresthesia (66)	2.7 ± 1.0	1.1 ± 1.2	<0.01
Respiratory and Otorhinolaryngological Symptoms			
Nasal obstruction (21)	3.0 ± 0.9	1.8 ± 1.1	<0.01
Sneezing (17)	2.1 ± 1.1	1.3 ± 1.1	<0.01
Cough (14)	2.7 ± 0.8	1.5 ± 1.2	<0.01
Headache (13)	2.5 ± 1.1	1.2 ± 1.3	<0.01
Foreign body sensation (8)	2.8 ± 1.0	1.5 ± 1.4	0.02
Nasal pruritus (7)	2.3 ± 1.4	0.9 ± 1.1	0.04
Hyposmia (6)	3.2 ± 0.8	2.7 ± 1.0	0.36
Cacosmia (5)	2.8 ± 1.1	1.2 ± 1.6	0.06
Pharyngodynia (1)	1.0	0.5	-
Dysphonia (1)	3.0	0.0	-
Eustachian tube catarrh (1)	4.0	1.0	-
Dyspnea (1)	1.0	1.0	-
Dysphagia (1)	3.0	0.0	-

**Table 3 diseases-12-00232-t003:** Distribution of DASS-21 categories before and after spa therapy (n = 144).

	Mean Total Scores ± SD			% (n)			
DASS-21 Categories (n)	Before Spa	After Spa	*p* Value *	Before Spa	After Spa	*p* Value *#*	Δ%
DEPRESSION							+45%
Normal (67)	5.0 ± 3.0	3.0 ± 4.0	<0.01	46.5 (67)	67.4 (97)	<0.01	
Mild (21)	11 ± 1.0	7.4 ± 4.5	<0.02	14.6 (21)	18 (26)	0.43	
Moderate (38)	17 ± 2.2	8.9 ± 5.9	<0.01	26.4 (38)	9 (13)	<0.01	
Severe/Extreme (18)	35 ± 10.7	17.7 ± 11.2	<0.02	12.5 (18)	5.6 (8)	0.04	
ANXIETY							+74%
Normal (62)	3.4 ± 2.3	2.7 ± 3.2	0.03	43.1 (62)	75.7 (109)	<0.01	
Mild (15)	8.0 ± 0.1	4.5 ± 3.1	<0.02	10.4 (15)	7 (10)	0.30	
Moderate (33)	11.3 ± 1.2	4.8 ± 5.0	<0.01	22.9 (33)	8.3 (12)	<0.01	
Severe/Extreme (34)	23.9 ± 10.9	10.8 ± 10.6	<0.01	23.6 (34)	9.0 (13)	<0.01	
STRESS							+63%
Normal (73)	8.4 ± 4.1	6.7 ± 6.0	<0.01	50.7 (73)	82.6 (119)	<0.01	
Mild (18)	16.8 ± 1.0	9.8 ± 5.5	<0.02	12.5 (18)	5.6 (8)	0.04	
Moderate (32)	21.8 ± 1.8	10.9 ± 8.2	<0.01	22.2 (32)	6.9 (10)	<0.01	
Severe/Extreme (21)	35 ± 8.9	15.4 ± 10.8	<0.02	14.6 (21)	4.9 (7)	<0.01	

* Wilcoxon test. # Chi-square test.

**Table 4 diseases-12-00232-t004:** Distribution of ISI categories before and after spa therapy (n = 39).

	Mean Total Scores ± SD		
ISI Categories (n)	Before Spa	After Spa	*p* Value
No clinically significant insomnia (36)	3.4 ± 2.2	3.4 ± 2.2	>0.05
Sub-threshold insomnia (33.3)	11.2 ± 2.2	7.1 ± 4.2	<0.05
Moderate insomnia (23)	17.7 ± 2.1	6.3 ± 3.2	<0.02
Severe insomnia (7.7)	24.0 ± 1.7	9.0 ± 6.6	-

**Table 5 diseases-12-00232-t005:** Distribution of DASS-21 (n = 144) and ISI (n = 39) categories before and after spa therapy stratified by sex.

DASS-21 Categories									
	Females (n = 98)				Males (n = 46)				
	Before Spa	After Spa	Δ%	*p* Value *	Before Spa	After Spa	Δ%	*p* Value *	*p* Value
Depression	13.1 ± 10.6	6.5 ± 6.1	−50%	<0.01	12.1 ± 11.1	8.8 ± 9.3	−27%	0.01	0.50 # 0.33 $
Anxiety	10.6 ± 10.2	4.7 ± 6.5	−57%	<0.01	10.4 ± 9.0	6.4 ± 7.4	−38%	<0.01	0.94 # 0.35 $
Stress	15.9 ± 10.4	8.5 ± 6.8	−47%	<0.01	17.3 ± 10.9	11.0 ± 9.7	−36%	<0.01	0.31 # 0.18 $
**ISI Categories**									
	**Females (n = 25)**				**Males (n = 14)**				
	**Before Spa**	**After Spa**	**Δ%**	***p* value ***	**Before Spa**	**After Spa**	**Δ%**	***p* value ***	***p* value**
Total mean score	13.4 ± 6.7	6.0 ± 3.9	−55%	0.01	6.3 ± 5.2	4.5 ± 4.3	−29%	>0.05	<0.05 # >0.05 $

* Wilcoxon test before vs. after spa therapy in females or males. # Mann–Whitney test before spa, females vs. males; $ Mann–Whitney test after spa, females vs. males.

**Table 6 diseases-12-00232-t006:** Impact of spa therapy on chronic disease-related symptoms stratified by previous SARS-CoV-2 infection.

	Without COVID-19 History				With COVID-19 History			
	n = 37				n = 107			
Symptoms, (n)	N	Mean VAS ± SD		*p* Value	N	Mean VAS ± SD		*p* Value
		Before Spa	After Spa			Before Spa	After Spa	
Arthro-rheumatic								
Pain in functional performance	26	3.3 ± 0.7	1.8 ± 1.2	<0.01	67	3.4 ± 0.5	2.1 ± 1.0	<0.01
Morning stiffness	25	3.2 ± 0.6	1.4 ± 1.3	<0.01	60	3.2 ± 0.8	1.8 ± 1.31	<0.01
Paresthesia	18	2.5 ± 1.0	1.0 ± 1.1	<0.01	48	2.8 ± 1.0	1.1 ± 1.3	<0.01
Respiratory								
Nasal obstruction	7	2.4 ± 1.1	1.1 ± 1.1	<0.01	14	3.2 ± 0.6	2.1 ± 0.9	<0.01
Sneezing	5	1.6 ± 0.9	1.0 ± 1.2	0.21	12	2.3 ± 1.1	1.4 ± 1.1	<0.01
Cough	4	5.5 ± 3.1	2.8 ± 1.6	0.30	10	2.6 ± 0.8	1.3 ± 1.3	0.01
Headache	4	2.8 ± 1.3	1.0 ± 1.4	0.13	9	2.4 ± 1.0	1.2 ± 1.3	<0.01
Foreign body sensation	4	2.5 ± 1.3	1.8 ± 1.5	0.32	4	3.0 ± 0.8	1.3 ± 1.5	0.04
Nasal pruritus	2	2.0 ± 1.4	0.5 ± 0.7	0.20	5	2.4 ± 1.5	1.0 ± 1.2	0.13
Hyposmia	3	3.0 ± 1.0	3.0 ± 1.0	1.00	3	3.3 ± 0.6	2.3 ± 1.2	0.42
Cacosmia	2	3.5 ± 0.7	1.5 ± 2.1	0.30	3	2.3 ± 1.2	1.0 ± 1.7	0.27
Pharyngodynia	1	1.0	0.0	NA	1	1.0	1.0	-
Dysphonia	0	-	-	-	1	3.0	0.0	-
Eustachian tube catarrh	0	-	-	-	1	4.0	1.0	-
Dyspnea	0	-	-	-	1	1.0	1.0	-
Dysphagia	0	-	-	-	1	3.0	0.0	-

**Table 7 diseases-12-00232-t007:** Distribution of DASS-21 (n = 144) and ISI (n = 39) categories before and after spa therapy stratified by previous SARS-CoV-2 infection.

DASS-21 Categories									
	with COVID-19 History (n = 107)				without COVID-19 History (n = 37)				
	Before Spa	After Spa	Δ%	*p* Value *	Before Spa	After Spa	Δ%	*p* Value *	*p* Value
Depression	13.6 ± 11.4	7.9 ± 8.0	−42%	<0.01	10.2 ± 8.3	5.2 ± 5.0	−49%	<0.01	0.17 # 0.08 $
Anxiety	11.3 ± 10.7	5.7 ± 7.3	−50%	<0.01	8.3 ± 6.1	4.0 ± 5.2	−52%	<0.01	0.25 # 0.18 $
Stress	17.1 ± 10.4	10.1 ± 7.7	−41%	<0.01	13.9 ± 10.5	7.0 ± 8.0	−50%	<0.01	0.10 # 0.008 $
**ISI Categories**									
	**With COVID-19 history (n = 29)**				**Without COVID-19 history (n = 10)**				
	**Before Spa**	**After Spa**	**Δ%**	***p* Value ***	**Before Spa**	**After Spa**	**Δ%**	***p* Value ***	***p* Value**
Total mean score	12.1 ± 7.2	5.4 ± 4.1	−55%	<0.01	7.4 ± 5.4	5.5 ± 4.2	−26%	>0.05	0.07 # 0.75 $

* Wilcoxon test before vs. after spa therapy in females or males. # Mann–Whitney test before spa, with COVID-19 history vs. without history; $ Mann–Whitney test after spa, with COVID-19 history vs. without history.

## Data Availability

Data are contained within this article.

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
