# Peer review of "Spa Therapy Efficacy in Mental Health and Sleep Quality Disorders in Patients with a History of COVID-19: A Comparative Study"

_diseases, 2024, doi:10.3390/diseases12100232_

Round 1
Reviewer 1 Report
Comments and Suggestions for Authors
The study appears interesting and encouraging for the use of a therapy that the ancient Greeks and Romans already considered effective.
There are two important considerations and comments to make:
1. the population studied is too heterogeneous therefore the problems that each group of patients presents cannot be simply evaluated with the tests described in the study even if their statistical validity is confirmed. Rheumatic problems cannot be compared with respiratory problems. The only common denominator could be the condition of inflammation, which has not been described
2. the subjects with previous SARS Cov 2 infection are not described based on the contracted form therefore they cannot be considered an eligible study group.
Author Response
Comments and Suggestions for Authors
The study appears interesting and encouraging for the use of a therapy that the ancient Greeks and Romans already considered effective.
There are two important considerations and comments to make:
Comment 1. The population studied is too heterogeneous therefore the problems that each group of patients presents cannot be simply evaluated with the tests described in the study even if their statistical validity is confirmed. Rheumatic problems cannot be compared with respiratory problems. The only common denominator could be the condition of inflammation, which has not been described
Response to Comment 1. We thank the Reviewer for this insightful feedback and for highlighting the importance of comparability between groups in our study, that are quite heterogeneous. Therefore, according to this Reviewer’s comments, we have revised the introduction to better explain that we have included patients that share an underlying common inflammatory condition, despite they have different clinical manifestations and diseases.
On page 2, lines 45-77, the following text was added “Since Romans, Spa therapies have been used for treating various medical conditions because of its healing, anti-inflammatory, antioxidant, and relaxing properties [8-11]. Indeed, inflammation is one of the principal pathophysiological mechanisms in multiple chronic conditions and can be targeted by Spa interventions, that can modulate inflammatory responses and reduce oxidative stress. For example, inflammation during airway infections and recurrence of infectious airway diseases are significantly decreased after Spa inhalation treatment [12-14], as well as tumor necrosis factor alpha (TNFα) and calprotectin levels in nasal mucosa [15], or interleukin(IL)-1 and IL-6 levels in nasal washes chronic obstructive pulmonary disease [11,16]. Moreover, IL-2, IL-4, and interferon-gamma (IFN-γ) during upper respiratory tract infections are reduced after Spa inhalation treatment, while anti-inflammatory cytokines, such as IL-10, are increased [16-18]. Spa inhalation therapies induce amelioration of inflammation and mucolytic actions, leading to improvements in normal acoustic curves functions with enhanced hearing in chronic upper airway and middle ear inflammation [19], or restoration of altered olfactory function in subjects with chronic rhinosinusitis [20]. After Spa balneotherapy or mud-balneotherapy, circulating prostaglandin E2, leukotriene B4, and TNFα are also reduced in patients with osteoarthritis or fibromyalgia [21-24], as well as redox status after Spa therapy in musculoskeletal [25,26] and skin disorders [27]. Despite these well-known beneficial effects of Spa therapy in inflammatory conditions, research on its effectiveness for mental health disorder treatment, such as anxiety, depression and sleep disorders, are limited. Therefore, a heterogeneous study population with different types of chronic inflammatory conditions, such as arthro-rheumatic, respiratory, and otorhinolaryngological diseases, were included in our investigation, to assess efficacy of Spa treatments on mitigating inflammation, regard-less underlying diseases, and to explore cross-sectional effects on various conditions.
In this study, we aim to examine efficacy of Spa therapy for well-being and mental health recovery in subjects with chronic inflammatory conditions and in COVID-19 survivors, thus proposing Spa treatments as a complementary and effective tool for improving mental health of this vulnerable population. Our results were also stratified by gender to identify different impacts on specific dimensions of psychological well-being, including anxiety, depression, and stress, and sleep disorders, as these aspects are often altered in individuals with chronic conditions, and Spa therapy can re-lief stress and improve mood and overall mental health.”
Comment 2. The subjects with previous SARS Cov 2 infection are not described based on the contracted form therefore they cannot be considered an eligible study group.
Response to Comment 2. In our study, we stratified subjects based on their history of COVID-19 infection, and most patients (n=107) had COVID-19 confirmed by molecular testing, while remaining subjects (n=37) had never been infected. Spa therapy significantly improved symptoms in both groups, with reduced anxiety, depression, and stress (p<0.01). Although improvements were more pronounced in subjects who had never contracted COVID-19, individuals with a history of SARS-CoV-2 infection exhibited higher baseline scores for anxiety, stress, and depression, reflecting its persistent impact on mental health.
For sleep disorders, those subjects with a previous SARS-CoV-2 infection showed a higher baseline ISI score (mean 12.1 ± 7.2, categorized as “sub-threshold insomnia”), that significantly reduced to “no clinically significant insomnia” after treatment (p<0.01). Conversely, never infected displayed no sleep disorders at baseline and post-treatment. Despite we did not further differentiate by severity or type of COVID-19 infection, our findings indicated a clear stratification between subjects based on infection history, and highlighted differential impact of Spa therapy on mental health outcomes.
Reviewer 2 Report
Comments and Suggestions for Authors
This paper seems to be product of circumstance, and while the results can be useful, the lack of a theoretical model underpinning the work (and providing new context for the results) is a little disappointing. There is a significant amount of research showing spa and related therapies improves one's mental state and healthful feelings (Cleveland Clinic has a site devoted to this). It is unclear to me why one would expect a different result based on the maladies focused on in this paper. There is where a model would be helpful. It would also be useful to understand how this is not related to participant confirmatory bias or Hawthorne Effect. Indeed, the study limitations laid out on page 8 (line 264+) are pretty substantial, making it difficult to generalize the results beyond those who took part in this study.
The paper would benefit from specifics and/or operational definitions. For example, line 54 - "Spa therapy on specific dimensions of psychological well-being." What are those specific dimensions? How to they connect with this topic and treatment? A deeper literature review on research looking at spa therapies and their acute and chronic / long-term impact on psychological health, mood and physical well-being is justified here too, again to be able to put this work in context.
Regarding study design, the first line (58) is confusing as I think you are putting too much information in one place. Maybe a table would help. It would be useful to understand how the different disorders were determined for the study (for example, why insomnia and not RLS or apnea?), and how the participants were recruit, and their data protected and verified. This looks like a sample of convenience, and if so, it is useful to lay out the limitations up front. A timeline showing the entire protocol would be helpful to so as to understand how all the pieces fit together. You could use this approach to show what a "spa cycle" looks like.
A power analysis is always helpful to put the statistical results in context. And again, more on the sample and its representativeness is needed. Specifics such as what is meant by "line 133- Following the spa cycle" would be helpful - is that immediately? For some period of time afterwards? If soon after, what data or theory do you have for a longer-lasting change, or the rate of spa therapy to maintain the beneficial effects. Do you expect that the effects will lessen over time, or the spa visits to be repeated more frequently - why or why not? Showing pathways for the impact on psychological as well as physiological health would be useful.
Lastly, I am concerned about statements such as: line 218 "Spa mineral waters have anti-inflammatory, analgesic... properties...". Has there been an analysis of the particular waters? Is it about the property of the waters or the warmth or the environment or some combination that makes the therapy as effective as it is, so something else. Without a model, I am left wondering how repeatable the study is and what it means practically. Also, the statement that females are more likely to suffer mental health disorders (line 242) is substantiated with one study, though it is a bold claim. There are different types of mental health issues for different reasons; I feel this claim needs to be qualified.
Author Response
Comments and Suggestions for Authors
This paper seems to be product of circumstance, and while the results can be useful, the lack of a theoretical model underpinning the work (and providing new context for the results) is a little disappointing.
Comment 1. There is a significant amount of research showing spa and related therapies improves one's mental state and healthful feelings (Cleveland Clinic has a site devoted to this). It is unclear to me why one would expect a different result based on the maladies focused on in this paper. There is where a model would be helpful. It would also be useful to understand how this is not related to participant confirmatory bias or Hawthorne Effect. Indeed, the study limitations laid out on page 8 (line 264+) are pretty substantial, making it difficult to generalize the results beyond those who took part in this study.
Response to Comment 1. We thank the Reviewer for these constructive comments, and we have addressed them as follows.
1A. Study Design and Environment: Our study was conducted in a controlled environment with consistent procedures for all participants. We aimed to minimize any external influences by standardizing Spa therapy and evaluation process.
1 B. Review of Cleveland Clinic Resources: On Cleveland Clinic website, we found the following site https://my.clevelandclinic.org/health/treatments/23137-hydrotherapy related to hydrotherapy, defined as “any method that uses water to treat a variety of symptoms throughout your body. You might see it called water therapy, aquatic therapy, pool therapy or balneotherapy”. Spa therapy not only includes balneotherapy, while also mud-baths and inhalation treatments, and waters used have different compositions of ions and other minerals, that give them therapeutic properties.
Therefore, we did not find specific studies or articles specifically related to benefits of Spa therapy in mental health conditions as described in our work.
1 C. Literature Review and Contextualization: We have expanded our literature review to include studies from other sources to better explore benefits of Spa therapies, integrated in introduction and discussion sections.
1 D. Hawthorne Effect Considerations: Despite Hawthorne Effect can influence participant behavior in some studies, we did not observe significant changes in participants' behavior, also because participants were primarily focused on their therapy and not on the study process itself. Indeed, improvements in psychological well-being and sleep quality were consistent across different groups, suggesting that these changes were more likely related to Spa therapy rather than participants' awareness of being studied.
Comment 2. The paper would benefit from specifics and/or operational definitions. For example, line 54 - "Spa therapy on specific dimensions of psychological well-being." What are those specific dimensions? How to they connect with this topic and treatment? A deeper literature review on research looking at spa therapies and their acute and chronic / long-term impact on psychological health, mood and physical well-being is justified here too, again to be able to put this work in context.
Response to Comment 2. We thank the Reviewer for this helpful comment that has improved the clarity of our manuscript.
2 A. Specific Dimensions of Psychological Well-Being: We have clarified specific dimensions of psychological well-being evaluated in our study, including anxiety, depression, and stress, as measured by the DASS-21 scale, as well as sleep quality assessed by ISI scale. These dimensions were chosen because they are commonly affected in patients with chronic conditions and are relevant to therapeutic effects of Spa therapy.
On page 2, lines 73-77, the following modifications were added “Our results were also stratified by gender to identify different impacts on specific dimensions of psychological well-being, including anxiety, depression, and stress, and sleep disorders, as these aspects are often altered in individuals with chronic conditions, and Spa therapy can relief stress and improve mood and overall mental health.”
2 B. Connection to Topic and Treatment: To better connect these dimensions to the context of Spa therapy, we have added a more detailed explanation in the introduction and discussion sections to make these connections clearer. On page 9, lines 295-298, the following text was added “These effects, especially relaxing properties, could also explain clinical improvements in psychological well-being by muscle tension and stress levels relief during sleep disorders and anxiety [56], further supporting Spa therapy as a complementary treatment for chronic conditions.”
Comment 3. Regarding study design, the first line (58) is confusing as I think you are putting too much information in one place. Maybe a table would help. It would be useful to understand how the different disorders were determined for the study (for example, why insomnia and not RLS or apnea?), and how the participants were recruit, and their data protected and verified. This looks like a sample of convenience, and if so, it is useful to lay out the limitations up front. A timeline showing the entire protocol would be helpful to so as to understand how all the pieces fit together. You could use this approach to show what a "spa cycle" looks like.
Response to Comment 3. We thank the Reviewer for these comments, and we have addressed the concerns as follows.
Study Design and Table: We agreed with the Reviewer, and we have added a new Table 1, summarizing the study design, including inclusion and exclusion criteria, treatment modalities, and data collection procedures. Please find below the new Table 1.
Table 1. Study design and population.
|
Feature |
Description |
|
Type |
Prospective observational study |
|
Participants |
144 Caucasian subjects |
|
Sex |
68% female, 32% male |
|
Age (Mean ± SD) |
56 ± 15 years (range: 24-78 years) |
|
BMI (Mean ± SD) |
25 ± 4.0 kg/m² |
|
Inclusion criteria |
Age ≥ 18 years Diagnosis of chronic arthro-rheumatic, respiratory, and/or otorhinolaryngological diseases Written informed consent |
|
Exclusion criteria |
Acute clinical conditions Cancer and autoimmune diseases Absence of signed written informed consent |
|
Diseases, n (%) |
Chronic arthro-rheumatic diseases, 106 (73.6) Respiratory diseases, 31 (21.5) Otorhinolaryngological diseases, 7(4.9) |
|
Duration |
2-week Spa cycle (balneotherapy and/or inhalation therapy) |
|
Protocols |
Balneotherapy Full-body immersion (excluding head) in Spa mineral waters at 37–38°C for 20 min. 30 min rest after treatment. 12-day complete cycle, consisting of 12 consecutive baths (once daily) in a single tub, preferably in the morning and under fasting conditions. For chronic arthro-rheumatic diseases.
Inhalation Crenotherapy Twelve 10-min applications of Spa mineral waters as direct jet and 12 aerosols. 10-min rest between applications. For chronic inflammation and/or irritation of upper and lower respiratory tracts and otolaryngological diseases. |
|
Outcome measures |
Symptoms: Visual Analogue Scale (VAS). Psychological well-being: Depression Anxiety Stress Scales-21 items (DASS-21) Sleep quality: Insomnia Severity Index (ISI). |
|
Data collection |
At baseline and after 2 weeks of Spa therapy |
On page 2, line 79-90, the paragraph has been rephrased as follows
“2.1 Population and Study Design
A total of 144 Caucasian subjects were included in this study and enrolled from three Italian Spas: “Stufe di Nerone” Baths (Bacoli – Naples), “Giardini Poseidon” Baths (Ischia – Naples), and “Terme dei Colli Asolani” Baths (Pieve del Grappa, Crespano, Treviso). These subjects had a diagnosis of chronic arthro-rheumatic, and/or respiratory, and/or otorhinolaryngological diseases and received a prescription from a general practitioner or specialist for a 2-week therapeutic Spa cycle (balneotherapy and/or inhalation). This study was approved by our local Ethics Committee “Campania Sud”, Naples, Italy (approval n. 44 r.p.s.o./2023 on March 7, 2023) and conducted in accordance with the Declaration of Helsinki. A specific case report form (CRF) was used to collect patients’ data. Participant demographics, inclusion and exclusion criteria, therapeutic interventions, treatment protocols, outcome measures, and data collection time points are summarized in Table 1.”
Comment 4. A power analysis is always helpful to put the statistical results in context.
Response to Comment 4. We thank the Reviewer for this observation, and missing analysis was included.
On page 4, lines 131-134, the following text was added “Post-hoc power analysis was performed using G*Power software (v.3.1.9.6; Franz Faul, Universitat Kiel, Germany), and for before-after Spa therapy, using the total cohort of 144 patients, power was 98%, while for comparisons between females (n=98) and males (n=46), power was 87%.”
Comment 5. And again, more on the sample and its representativeness is needed. Specifics such as what is meant by "line 133- Following the spa cycle" would be helpful - is that immediately? For some period of time afterwards? If soon after, what data or theory do you have for a longer-lasting change, or the rate of spa therapy to maintain the beneficial effects. Do you expect that the effects will lessen over time, or the spa visits to be repeated more frequently - why or why not? Showing pathways for the impact on psychological as well as physiological health would be useful.
Response to Comment 5. We thank the Reviewer for these comments, and outcomes were measured on day 0 (day at enrollment before starting Spa cycle) and at the 12th day of Spa cycle at the end of treatment. We have previously documented that improvements observed immediately after the end of Spa cycle tend to diminish over time, while never returning to basal pre-Spa conditions (Costantino M.; Lampa E. Long-Term effects of sulphurous well mud-therapy: epidemiological study. Med Clin Term 2002, 14(50-51),347-361); however, if patients consistently undergo at least one therapeutic Spa cycle per year, a progressive accumulation of therapeutic effects have been observed due to annual continuity of Spa therapy.
Comment 6. Lastly, I am concerned about statements such as: line 218 "Spa mineral waters have anti-inflammatory, analgesic... properties...". Has there been an analysis of the particular waters? Is it about the property of the waters or the warmth or the environment or some combination that makes the therapy as effective as it is, so something else. Without a model, I am left wondering how repeatable the study is and what it means practically. Also, the statement that females are more likely to suffer mental health disorders (line 242) is substantiated with one study, though it is a bold claim. There are different types of mental health issues for different reasons; I feel this claim needs to be qualified.
Response to Comment 6. We thank the Reviewer for this point, and we have extended literature review for both observations.
On page 8, lines 241-245, the following text was added “Spa mineral waters have anti-inflammatory, analgesic, and decongestant properties, as well as relaxing environment and positive social interaction, as already described in previous in vitro studies and clinical trials, like demonstration of antioxidant effects of sulfurous/bicarbonate/sodic Portuguese Spa mineral water mediated by increased inducible nitric oxide synthase expression [37-40].”
On page 9, lines 272-275, the following text was added “Females are more likely to suffer from mental health disorders compared to males, because of a higher burden and an increased susceptibility, with a global annual prevalence of depression of 5.5% in females and 3.2% in males, with a 1.7-fold greater incidence in women, as also documented by several World Health Organization’s reports [52-55].”
The following references were also added.
- Melgar-Sánchez, L.M.; García-Ruiz, I.; Pardo-Marqués, V.; Agulló-Ortuño, M.T.; Martínez-Galán, I. Influence of mineral wa-ters on in vitro proliferation, antioxidant response and cytokine production in a human lung fibroblasts cell line. Int J Biome-teorol. 2019 Sep;63(9):1171-1180. doi: 10.1007/s00484-019-01730-0. Epub 2019 Jun 21. PMID: 31227887.
- Silva, A.; Oliveira, A.S.; Vaz, C.V.; Correia, S.; Ferreira, R.; Breitenfeld, L.; Martinez-de-Oliveira, J.; Palmeira-de-Oliveira, R.; Pereira, C.M.F.; Palmeira-de-Oliveira, A.; Cruz, MT. Anti-inflammatory potential of Portuguese thermal waters. Sci Rep. 2020 Dec 18;10(1):22313. doi: 10.1038/s41598-020-79394-9. PMID: 33339881; PMCID: PMC7749128.
- Yurtkuran, M.; Yurtkuran, M.; Alp, A.; Nasircilar, A.; Bingöl, U.; Altan, L.; Sarpdere, G. Balneotherapy and tap water therapy in the treatment of knee osteoarthritis. Rheumatol Int. 2006 Nov;27(1):19-27. doi: 10.1007/s00296-006-0158-8. Epub 2006 Jul 11. PMID: 16832639.
- Carbajo, J.M.; Maraver, F. Sulphurous Mineral Waters: New Applications for Health. Evid Based Complement Alternat Med. 2017;2017:8034084. doi: 10.1155/2017/8034084. Epub 2017 Apr 6. PMID: 28484507; PMCID: PMC5397653.
- Albert, P.R. Why is depression more prevalent in women? J Psychiatry Neurosci. 2015 Jul;40(4):219-21. doi: 10.1503/jpn.150205. PMID: 26107348; PMCID: PMC4478054.
- Srivastava, K. Women and mental health: Psychosocial perspective. Ind Psychiatry J. 2012 Jan;21(1):1-3. doi: 10.4103/0972-6748.110938. PMID: 23766570; PMCID: PMC3678171.
Reviewer 3 Report
Comments and Suggestions for Authors
Table 4 depression in females after spa is incorrect
Discussion: Limitations are more than mentioned.Lack of a control group does not only limit generalization, but makes it also not possibleto judge, whether the effects of the spa.therapy are specific to the treatment.
Conclusion: Although you mention (at elast patially) in the dicsussion that the study i only observationela, and that there is no contorol group you state that "Spa therapy can be considered an effective complementary therapeutic option"
The conclusions should be more moderate regarding that the study design is observational without control group.
Otherwise your article is quite well written.
Author Response
Comments and Suggestions for Authors
Comment 1. Table 4 depression in females after spa is incorrect.
Response to Comment 1. We apologize for the mistake in Table 4 (now Table 5), and we thank the Reviewer for pointing out this error. We have corrected and updated this table accordingly.
Comment 2. Discussion: Limitations are more than mentioned. Lack of a control group does not only limit generalization, but makes it also not possible to judge, whether the effects of the spa therapy are specific to the treatment.
Response to Comment 2. We agreed with the Reviewer that the absence of a control group limits generalizability of our findings, and we have included this point in the limitations section.
On page 9, lines 299-304, the paragraph has been rephrased as follows “Our study has several limitations: (i) its observational nature that did not allow to establish a direct causal relationship between Spa therapy and observed improvements; (ii) absence of a control group; and (iii) small sample size for certain categories that did not permit specific comparisons, such as those related to insomnia severity categories. Therefore, further research with controlled and randomized designs is needed to confirm these findings.”
Comment 3. Conclusion: Although you mention (at least partially) in the discussion that the study in only observational, and that there is no control group you state that "Spa therapy can be considered an effective complementary therapeutic option"
The conclusions should be more moderate regarding that the study design is observational without control group.
Otherwise, your article is quite well written.
Response to Comment 3. We have revised the conclusion section to taper our sentences, considering the observational nature of our study and the absence of a control group.
On page 9, lines 306-310, conclusions have been adapted as follows “In conclusion, our findings suggested that Spa therapy could be a valuable complementary therapeutic option for improving psychological well-being and sleep quality in patients with chronic conditions, including those with a history of COVID-19 infection. However, further controlled and randomized studies are needed to confirm these findings.”
We would like to thank the Reviewer for this positive feedback and for these helpful comments that have markedly improved the quality of our work.
Round 2
Reviewer 1 Report
Comments and Suggestions for Authors
Lines 32-34 ……COVID-19 pandemic has posed an unprecedented challenge to global public health, 33 and beyond its the direct physical effects, its long-term impact on mental health and psychological well-being of affected individuals is becoming increasingly evident .
if the disease is Covid 19 + the main topic, it is necessary to specify what type of infection was contracted, what course the disease had, whether the patient was hospitalized or not
In lines 67-70 the covid disease is not indicated and specified.
It must be specified what type of severity the patients presented. If this is not possible, it must be stated that the patients have contracted Covid 19 and have carried out the therapy at home. Having contracted Covid 19 has aggravated the previous illness to what extent? Unfortunately this remains a large gap and it is difficult to evaluate the study correctly.
Author Response
Reviewer: 1
Comments and Suggestions for Authors
Comment 1. Lines 32-34 ……COVID-19 pandemic has posed an unprecedented challenge to global public health, 33 and beyond its the direct physical effects, its long-term impact on mental health and psychological well-being of affected individuals is becoming increasingly evident . if the disease is Covid 19 + the main topic, it is necessary to specify what type of infection was contracted, what course the disease had, whether the patient was hospitalized or not.
Response to Comment 1. We thank the Reviewer for this valuable feedback and for pointing out the need for additional details regarding type and course of COVID-19 infection.
On page 6, lines 197-205, the paragraph was rephrased as follows “Finally, enrolled subjects were stratified by at least one previous SARS-CoV-2 infection, and most patients (n= 107) had contracted COVID-19 confirmed by molecular test (mean age ± SD, 54±15 years; range, 24-78 years; median age, 57 years; female 72% and male 28%; mean BMI [Kg/m 2 ] ± SD, 25±4.2). In details, 68% of them (n = 73) re-ported mild COVID-19 treated at home, while 32% (n = 34) had moderate/severe forms, with only one patient requiring oxygen support, although no subjects were admitted to intensive care units. Average symptom duration was 9±1.2 days for mild and 13±2.2 days for moderate/severe cases. Moreover, 65% (n=70) of those subjects who had a previous COVID-19 infection reported long-term sequelae.”
Comment 2. In lines 67-70 the covid disease is not indicated and specified. It must be specified what type of severity the patients presented. If this is not possible, it must be stated that the patients have contracted Covid 19 and have carried out the therapy at home. Having contracted Covid 19 has aggravated the previous illness to what extent? Unfortunately this remains a large gap and it is difficult to evaluate the study correctly.
Response to Comment 2. We apologize for this missing information, and we have clarified type and severity of COVID-19 infection in our study population. We have also addressed the potential impact of COVID-19 on pre-existing chronic inflammatory conditions, although it was not possible to quantify the exact extent for each individual condition. The revised text reflects this clarification and the focus of the study, which is on assessing the efficacy of Spa treatments in improving mental health and reducing inflammation, regardless of the specific underlying disease.
On page 2, lines 65-77, text was rephrased as follows “Therefore, a heterogeneous study population with different types of chronic inflammatory conditions, such as arthro-rheumatic, respiratory, and otorhinolaryngological diseases, as well as COVID-19 survivors, were included in our investigation, to assess efficacy of Spa treatments on mitigating inflammation, regardless underlying diseases, and to explore cross-sectional effects on various conditions.
In this study, we aim to examine efficacy of Spa therapy for well-being and mental health recovery in subjects with chronic inflammatory conditions and in COVID-19 survivors, thus proposing Spa treatments as a complementary and effective tool for improving mental health of this vulnerable population. Our results were also stratified by gender to identify different impacts on specific dimensions of psychological well-being, including anxiety, depression, and stress, and sleep disorders, as these aspects are often altered in individuals with chronic conditions, and Spa therapy can relief stress and improve mood and overall mental health.”
Reviewer 3 Report
Comments and Suggestions for Authors
Your observational study is well performed and the article well written.
I very much hope that in the future you will perform "further controlled and randomized studies which are needed to confirm these findings." because this is what balneology needs more than more observationela studies.
Author Response
Reviewer: 3
Comments and Suggestions for Authors
Your observational study is well performed and the article well written. I very much hope that in the future you will perform "further controlled and randomized studies which are needed to confirm these findings." because this is what balneology needs more than more observationela studies.
Response to Comment. We would like to thank this Reviewer for this very positive feedback and for recognizing the quality of our observational study. We appreciate your suggestion regarding the need for controlled and randomized studies to further validate our findings, and we agree that such studies would be valuable for advancing the field of balneology and confirming efficacy of Spa treatments. We are committed to continuing our research in this area and will certainly consider designing controlled and randomized trials in the future to increase our current knowledge in this field. Your feedback reinforces the importance of rigorous study designs, and we are excited about the prospect of contributing to this important field with more robust evidence.
Thank you again for your valuable insights and encouragement.
Round 3
Reviewer 1 Report
Comments and Suggestions for Authors
The corrections made to the text make the study easier to read and understand as well as more consistent with the characteristics of scientific eligibility.